# An Evaluation of Health Behavior Change Training for Health and Care Professionals in St. Helena

**DOI:** 10.3390/healthcare13040435

**Published:** 2025-02-18

**Authors:** Wendy Maltinsky, Vivien Swanson, Kamar Tanyan, Sarah Hotham

**Affiliations:** 1Psychology Division, School of Natural Sciences, University of Stirling, Stirling FK9 4LA, UK; vivien.swanson@stir.ac.uk; 2Health and Social Care Portfolio, Jamestown STHL 1ZZ, Saint Helena; kamar.tanyan@doctors.org.uk; 3Centre for Health Services Studies, University of Kent, Canterbury CT2 7NF, UK; s.hotham@kent.ac.uk

**Keywords:** health behavior change, health multi-professional training, low-middle-income country, evaluation

## Abstract

**Background:** Health behavior consultations support self-management if delivered by skilled practitioners. We summarize here the results of a collaborative training intervention program delivered to health and care practitioners working in a remote-island context. The program was designed to build confidence in the implementation of communication and behavior change skills and to sustain their use in work settings. The setting for the behavior change training program was the South Atlantic island of St. Helena, a remote low-middle-income country which has a population with high levels of obesity and a prevalence of long-term conditions. **Objectives:** We aimed to increase knowledge, confidence, and implementation of behavior change techniques (BCTs) and communication skills of health and social care staff through delivering and evaluating training using the MAP (Motivation, Action, Prompt) behavior change framework. A successful training intervention could ultimately improve self-management and patient health outcomes. **Methods:** Co-production with onsite representatives adapted the program for local delivery. A two-day training program was delivered face-to-face to 32 multidisciplinary staff. Pre- and post-intervention and 18-month follow-up evaluation assessed reactions, learning and implementation using multiple methods, including participant feedback and primary care patient reports. **Results:** Positive reactions to training and significant improvement in confidence, perceived importance, intention to use and implementation of BCTs and communication skills immediately post-training and at long-term follow-up were observed. Patient reports suggested some techniques became routinely used. Methodological difficulties arose due to staff retention and disruption through the COVID-19 pandemic. **Conclusions:** The delivery of health behavior change training can be effective in remote contexts with sustainable impacts on healthcare. There are challenges working in this context including staff continuity and technological reliability.

## 1. Introduction

Supporting people to improve health and prevent or manage long-term conditions (LTCs) is a global challenge. There are individual physical, financial, family and social costs to bear as a consequence of LTCs [1,2,3]. In the global south, the impact of LTCs is more profound than in the global north, with individual, community, workforce, and economic burdens being even greater [4]. Improving lifestyle and nutrition behaviors can help in the management and prevention of LTCs [1,2]. Positive health behaviors including nutritional consumption, sleep, physical activity and a reduction in sedentary behavior are key in determining health outcomes [5,6], but behavior change can be difficult to achieve and sustain [7]. Health and social care professionals (HCPs), including those working in primary and secondary care settings and wider health promotion staff, can use formal and informal consultation opportunities to instigate and support positive health behavior change conversations. Consultations incorporating evidence-based health behavior change techniques have optimal impact [8]. Evidence suggests that these types of “brief interventions” delivered by the health and social care workforce can have a positive impact on alcohol consumption [9], physical activity [10], smoking behavior [11], and short-term changes in dietary behaviors [12]. Brief interventions have also shown promise in preventing the onset of LTCs [13].

For these conversations to be effective clinicians require competence and confidence in using communication skills and behavior change techniques [14]. Communication skills such as active listening, asking open questions and being empathetic are important as part of health behavior change conversations [15,16]. For clinicians to use these skills with confidence, they generally require training [17]. Mentoring can assist in sustaining the skills developed through training [18].

This study reports on a collaborative training intervention program for health and social care practitioners working in a remote-island context. It was designed to build confidence in the implementation of these skills and to sustain their use in work settings.

The island of St Helena in the South Atlantic Ocean is an Official Developing Countries Aid recipient, an upper-middle-income country and a British Overseas Territory. It has a population of between four and five thousand people, and an aging population with a falling birth rate [19]. Due to its size and isolation, the island provides a unique, small, closed environment. There are significant health problems, including high levels of obesity and type 2 diabetes [20]. The 2018–19 St Helena Government Health Promotion Strategy [20] adopted a community-wide approach to address these problems and to bring about change to general population health. The strategy was designed to incorporate primary, secondary and tertiary services addressing the prevention of the onset of disease/ill health and improved self-management of disease. The approach was considered favorable to bring about a systemic change that would ultimately cascade across the age range to foster positive long-term healthy habits. Changing behaviors such as sedentary behavior, smoking and diet was a key focus of the strategy. This presented a clear need to train HCPs in evidence-based behavior change techniques (BCTs) [21] alongside communication skills to enable these BCTs to be delivered competently in a consultation [22] to support the implementation of the strategy in work settings.

BCTs are replicable intervention components that can alter behavior [23] and are used individually or in combination in behavior change interventions. Consultations or conversations using BCTS can help people to change behavior and lead to improved health outcomes [24,25]. The “Health Behaviour Change Competency Framework” (HBCCF) is a theoretically informed organized description of BCTs [26] which categorizes them according to their function. It is important to note that the HBCCF and MAP categorization draws on the first version of the BCT Taxonomy (BCT Taxonomy V1) which identified 93 BCTs [23] now superseded by versions with greater numbers of identified BCTs [27]. The three functions of BCTs are “Motivation, Action and Prompt” (MAP) [28]. BCTs in the motivation category are those that assist people to build motivation using techniques such as: M10 (HBCCF) BCT 5.1 (BCT taxonomy V1); ”providing information about health consequences”. Those that facilitate bridging the gap between positive intention and behavioral enactment are in the Action category, such as A8 (HBCCF) BCT 1.4 (BCT taxonomy V1); ”creating an action plan”. Finally, those that can prompt or cue people to enact their behaviors are in the Prompt category, such as BCT 7.1 (BCT Taxonomy V1); a prompt might be placing a reminder for the behavior on one’s phone (P2, HBCCF). Generating a collaborative conversation using appropriate communication skills can be key to positive health outcomes [29] so these skills should be incorporated into consultation training where possible.

The MAP training program based on the HBCCF framework has been used successfully [30] to deliver training for staff supporting behavior change to improve health and wellbeing. It uses 10 BCTs from the HBCCF framework that have been selected based on their relevance and strong evidence base for the most commonly addressed health behaviors [8,31].

Using a multidisciplinary team approach in training can help to support more effective implementation of changes and adoption and embedding of training programs such as MAP in routine care. Evidence suggests that effective training for healthcare professionals should embrace collaborative team-based approaches that provide opportunities for social interaction and learning from others outside of individual specialisms [32]. Hence, using the MAP framework is an excellent “fit” for a context like St Helena, where holistic, collaborative multidisciplinary working is common in health and social care. Mentoring or coaching can help to sustain skills learned during training [18].

A partnership project between the University of Stirling and St Helena Public Health, supported by funding from the UK Burdett Trust for Nursing and Public Health England was developed to offer training for health and social care staff in the use of the MAP behavior change techniques and communication skills. The training objectives were to increase knowledge, confidence and implementation of these skills.

The overall project aims were to co-develop, deliver and assess, effectiveness, acceptability and sustainability of the health behavior change training program.

The key research questions were:Can a co-developed health behavior change training intervention be effective in increasing HCP use of BCTs and communication skills?What are HCP participant perceptions of the acceptability of the training interventionWill HCP training have an impact on patient perception of practitioner use of communication and BCT skills post training?Does mentor training help to sustain practitioner consultation skills longer-term?

## 2. Materials and Methods

### 2.1. Design

A longitudinal pre–post cohort design was used. The cohort included multidisciplinary staff working in healthcare-related roles in St. Helena who received the training intervention. Cross-sectional surveys of the views of the patients of training participants were also carried out at two time points; pre-intervention and at 18-month follow up. A self-nominated group of four participants from the multidisciplinary staff group was established as the mentoring group. They received mentoring training with the aim of providing ongoing local support in the use of the BCTs and communication skills to the rest of the trained cohort. Figure 1 illustrates the timeline and data collection time points and participant group.

### 2.2. Intervention Development

The tailored MAP program was developed by the research team and adapted in collaboration with stakeholders for in-person delivery in St Helena. The original program had been used to train health practitioners working in diabetes care [30], healthcare workers in Ethiopia [33] and more widely with staff across NHS Scotland. The adaptation and content was informed by discussion with Public Health England to ensure messages from pre-existing public health programs, including their Healthy Weight Strategy, were fully captured in the training. Adaptations were also made in the period prior to delivery through a series of online remote discussions with a steering group drawn from representatives across St Helena health and social care staff. This rendered the program contextually appropriate, ensuring it was focusing on relevant issues and the specific needs of the local population. For example, case study examples used during behavior change skills practice sessions referred to the current smoking cessation campaign and reflected the role of the extreme topography of St Helena in limiting accessibility of fresh fruit and vegetables, which made both physical activity and healthy eating more challenging in this context.

### 2.3. Intervention Content

The ”MAP” (motivation, action, prompts) individual behavior change training program formed the basis of the intervention. There were two broad areas of focus: (i) develop participants’ knowledge, competence and confidence in using communication skills needed for behavior change; (ii) develop participants’ knowledge of and competence in use of a set of evidence-based techniques for changing individual behaviors. Communication skills included active listening, summarizing and reflecting. The BCTs included two motivational techniques, four action techniques and four prompting techniques.

### 2.4. Delivery

The behavior change training program included 16 h of learning, delivered over a 2-week period in October 2019. It was delivered in person by 3 members of the research team in four half-day sessions, to two groups of staff (4 sessions per group) allocated randomly to either morning or afternoon sessions based on availability.

Delivery methods included a range of training and participatory techniques based on adult learning principles. These included PowerPoint presentations and small group discussion; skill rehearsal role plays with feedback; games and activities. We used observation and video and audio recordings to facilitate reflection on skill development.

To achieve sustainability, the program also aimed to develop skills in mentoring and coaching for a small number of HCPs (n = 4) who could act as future local ”champions” or ”mentors” to achieve sustainable impact. Two sessions of face-to-face mentoring training, including learning mentoring theory and mentoring skills practice, and one information session with the NHS Senior management team were also carried out.

### 2.5. Participants

Two groups of participants were recruited. One group was the practitioner recipients of the training and the second group was patients of those who were receiving the training. Figure 1 and Table 1 provide information on intervention and data collection time points alongside the measures used and the relevant participant group.

#### 2.5.1. Practitioner Training Participants

The training was developed to be appropriate for all health and social care-related staff in St Helena, and all staff on the island were eligible for inclusion in the training intervention. There were approximately 250 staff working in health and social care at the time of delivery. The health directorate was responsible for participant recruitment, taking into account the following: (1) the need to ensure that health and care settings would remain sufficiently staffed; (2) that there was broad representation across different health and care settings. The final selection of participants was also made in consultation with local clinical leads, on the basis of their roles providing opportunity for behavior change consultations (patient or public-facing) and willingness and availability to attend training. The practical need to minimize disruption to health and care services necessarily limited the numbers who could attend the training but nevertheless demonstrated the commitment of senior managers to the training program.

There were 32 HCP participants, 17 and 15 in each group; 8 (25%) were male. A wide range of job roles were represented in the training, including hospital and community nurses, doctors and healthcare assistants or support workers, pharmacists, social workers, dental care staff and environmental health officers as well as health administrators.

Most (21, 70%) had been in their current role for less than 10 years. The majority (21, 70%) saw up to 15 patients per week, although some were not in a patient-facing role but worked in health-promoting roles with colleagues or the general public. Many (18, 62%) had no previous training in behavior change; 9 (38%) had training in communication skills or brief interventions.

Follow-up: There were 14 responses (2 male, 11 female, 1 no response) from the original 32 practitioner participants at 18-month follow-up from staff, including doctors, community nurses, healthcare assistants, and local government officers.

#### 2.5.2. Patients of the Practitioner Training Recipients

Samples of patients were recruited from primary care health centers to take part in an anonymous pre–post intervention evaluation of health professionals’ communication and behavior change competencies. We did not specify criteria for inclusion, except that they were attending a consultation with a practitioner who was participating in the training.

### 2.6. Evaluation Measures

The impact of the training was evaluated with practitioner participants (3 time points) and patient recipients (2 time points) using a range of methods, following Kirkpatrick’s [34] hierarchical model of levels of assessment of the impact of training. The Kirkpatrick model allows us to understand the effectiveness of the training as well as its acceptability by the impact on behavior and learning. Sustainability is measured by establishing how enduring these changes are across time, measured at 18-month follow-up. Table 1 illustrates the measures, constructs, timescales and integration with the Kirkpatrick model. We covered the first three of four levels in our evaluation: the reaction of participants to the training; the learning of skills; and behavior, i.e., the implementation of training skills by participants. We were unable to evaluate the final level, results or impact on recipient health, due to resources and the length of the study.

Patients of practitioners who participated in the training completed questionnaires at two time points: (i) pre-intervention (3 weeks) and (ii) 18 months post-intervention. Pre-intervention patient recruitment (n = 25) was carried out by community nursing and reception staff in primary care (health center) settings. They were asked to hand out questionnaires opportunistically to adult practice attenders at random over a period of three weeks prior to the start of the set training time period. At 18-month follow-up, data were collected at the same centers by a nurse-researcher who approached patients post-consultation. Due to more resources becoming available, a larger sample (n = 74) was recruited at follow-up. For comparative analysis, we matched a sub-sample of n = 25 from this group to the initial group on the basis of age and sex criteria.

Follow-up data collection was planned for 3 months and 1 year post-delivery. However, due to the COVID pandemic, we were unable to collect follow-up data (for staff and patient groups) until 18 months post-delivery. We held one post-course mentoring session (remote delivery), but three members of the mentoring team also left the program during the pandemic.

#### 2.6.1. Reaction

Training usefulness and acceptability were obtained immediately post-training using an adapted Training Acceptability Rating Scale (TARS) [35]. This included 5 questions rated from 1 (low) to 4 (high) assessing understanding, skill development, confidence, usability, and satisfaction. The TARS questionnaire included three open, free-text responses. We included these as part of the feedback questionnaire because they could provide additional insight into the perceived value of the training, the parts of the training that were considered the most useful to the application of BCTs and communication skills in their practice, and any changes that would improve the training. Open questions of this nature can contribute to a deeper understanding [36], and in this case, we wanted to understand how and what parts of the training might assist in changing consultation practice.

#### 2.6.2. Learning and Behavior

##### Training Participant Ratings

1. BCT/Care Questionnaire: Participant self-rated measures of 5 communication skills and 10 BCTs (see below) were completed at 3 time points—Time 1: 3 weeks before, Time 2: immediately post-training, and Time 3: at long-term (18 m) follow-up. Behavior (skills use) was rated only at time points 1 and 3.

The 10 BCTs were assessed in relation to 4 factors: (1) confidence (how easy/difficult) (2) perceived importance of this technique for practice; (3) intention to use in future practice; (4) behavior (reported use) in patient consultations. The specific techniques included the following:

2. Five communication skills, adapted from the CARE (consultation and relational empathy) measure [37]. These were as follows: put the person at ease; let them tell their story; really listen; understand; empower (let them take control). These are conversation skills that are compatible with delivering effective behavior change consultations.

3. Ten MAP behavior change techniques (BCTs): Identify the behavior; discuss pros and cons; action plan; monitor behavior; access social support; problem solving; environmental cues; use reminders to prompt behavior; repetition; and review progress. These items were scored on a 4-point Likert scale from 1 = ”not at all” to 4 = “a great deal” and summed to create an overall score for each BCT component: confidence, importance, intention, and behavior. We also compared reported the behavior (frequency of use) of each of the 15 BCTs and communication skills at Time 1 and Time 3. It was not appropriate to measure behavior at the end of the workshop (Time 2) since participants had not had the opportunity to use their skills in practice. Intention to use can signal the motivation an individual has to use the skill [38,39]. Measuring this variable over and above the use of skills indicates when people may intend to use it but may not have had the opportunity to do so.

##### Patient Ratings

Health professionals’ behavior in relation to their use of skills in practice was also assessed by inviting a sample of primary care patients to rate practitioner behavior immediately after consultations. The patients were invited to complete a brief anonymous survey at 3 weeks pre-training (Time 1) and at 18-month follow-up (Time 3).

The pre–post patient questionnaire adapted the Picture Format 12 Item CARE measure [37] to include 5 questions about communication skills (empathic care) and 7 questions about practitioner use of BCTs delivered in the training intervention. They were asked to rate “today’s consultation” with the following root: “how was the health professional at…?” The 12 items were as follows: “making you feel at ease”; letting you tell your story; really listening; understanding your concerns; helping you take control; decide a specific behavior to change; work out pros and cons; solve barriers; help to monitor; set reminders; find social support; and review goals. These were scored from 1 (not at all) to 5 (a great deal), and total communication skills and behavior change skills were calculated separately.

### 2.7. Analysis

Analysis was carried out using Statistical Package for Social Science (SPPS) v27 software [40]. Total scores for participant communication skills and BCTs (confidence, importance, and future intention) were compared at Time 1, Time 2, and Time 3, and behavior at Time 1 and Time 3 only, using paired t-tests or repeated measures ANOVA tests with post hoc comparisons. Eta squared or Cohens d were used to estimate the effect size. The sample size was small. Small amounts of data were missing at random but amounted to less than 5% overall. Missing data were replaced with the series mean where appropriate.

Patients’ health professional rating data pre- and post-intervention were compared using independent t-tests (paired tests were not appropriate as different people participated at each time point). Bonferroni correction was applied to significance testing to reduce the likelihood of type 1 error due to multiple testing.

The three open questions in the TARS questionnaire were treated by following a content analysis approach [41]. For all of the questions, two of the researchers (WM and VS) familiarized themselves with the data and discussed possible categories. WM subsequently categorized the data, and VS reviewed. Any disagreements were resolved through discussion. The two questions asking about the ”most helpful” part and ”what changes could be made” to the training, were first categorized as negative or positive. All of the responses from the three questions were then combined to explore themes.

### 2.8. Ethical Permission

Approved by UK IRAS 30/7/2019; Reference 271155. Project title: Improving people centered care using behavior change—St Helena.

## 3. Results

The results are presented in relation to the Kirkpatrick model grouping measures relevant to ”reaction” to training; ”learning” and ”behavior”. Patient data were included under the heading of ”learning and behavior”.

### 3.1. Reaction to Training

Engagement: Most participants (n = 32) attended two full days (four sessions) of training, with a small minority (n = 4) missing random one-half-day session.

Course Evaluation: The TARS measure recorded positive mean scores (max 4 for each item) about aspects of the training, including understanding of concepts (mean 3.4, SD 0.7), developing skills (mean 3.4, SD 0.6), confidence (mean 3.1, SD 0.8), willingness to use skills (3.4, SD 0.8), and overall satisfaction with training (mean 3.5, SD 0.5).

Thirty (94%) participants responded to the three open-ended questions. Altogether, 66 responses were gathered across the three questions. For ”most helpful part” and ”any suggested changes”, the positive comments indicated that the interactive approaches were most welcomed. The use of ”role play” was cited most frequently as a helpful activity alongside group discussion activities. The only change recommended was fewer questionnaires. One person suggested the use of follow-on courses. The open responses were categorized into three themes, illustrated by the examples below:

Theme 1: Participants’ increased confidence:

“This training has and will help me in further chronic disease management clinics. Previously I had NO confidence in running these clinics. Now I feel better and more confident.”P22

Theme 2: Perceived value of collaborative consultations with patients:

“I am reflecting on myself as well as others. Understanding the perspective of pts and empowering pts to get the eureka! Moment, rather than dictate to them what they should be doing.”P4

Theme 3: The need to focus on small changes:

“Thinking about structuring a conversation before I have it realising that I don’t have to get it all into one session.”P24

### 3.2. Learning and Behavior

Positive changes in BCT and communication skills were recorded comparing participants’ questionnaire data before and after the training and at follow-up.

#### 3.2.1. Communication Skills

Figure 2 below presents summary data on confidence, importance, intention, and behavior in relation to the reported use of the five communication skills at three time points.

Confidence in skill use (t(23) = 2.35, *p* = 0.014, d = 0.47) and importance of skills (t(23) = 1.8, *p* = 0.04, d = 0.37) increased between baseline (pre-training) and the end of training. There was no change in intention to use skills (t(15) = 0.40, *p* = 0.36, d = 0.09).

Confidence significantly increased between baseline and follow-up (F(2,20) = 8.9, *p* = 0.002 eta^2^ = 0.47). There was no significant increase in importance (F(2,18) = 3.09, *p* = 0.07, eta^2^ = 0.25) or intention (F (2,8) = 1.54, *p* = 0.27, eta^2^ = 0.27).

Participants reported increased use of communication skills (behavior) between Time 1 (pre-intervention, mean 14.1 (SD 4.1) and Time 3 (follow up, mean 16.9 (SD4.4) (t (8) = 2.25, *p* = 0.03, d = 0.79).

#### 3.2.2. Behavior Change Techniques

Figure 3 below presents pre- and post-intervention and follow-up data for confidence, importance, intention and self-reported behavior for use of MAP behavior change techniques.

Confidence (t(22) = 5.1, *p* < 0.001; d = 1.0) and importance (t(20) = 3.8, *p* < 0.001; d = 1.0) significantly increased from Time 1 (pre-training) to Time 2 (post-training) but intention to use did not (t(12) = 1.4, *p* = 0.10; d = 0.4).

Confidence significantly increased from Time 1 to Time 3 (F (2,16) = 26.83, *p* < 0.001, eta^2^ = 0.8). Importance also significantly increased (F (2,18) = 6.3, *p* = 0.008, eta^2^ = 0.41), *p* < 0.001, whereas intention did not change (F(2,8) = 0.74, *p* = 0.50, eta^2^ = 0.16).

We investigated whether the use of specific BCTs in practice had changed between Time 1 (pre-training) and Time 3 (follow-up) using paired tests (see Figure 4). The reported use of all BCTs increased. Using self-monitoring (*p* = 0.003); social support (*p* < 0.001); problem solving (*p* < 0.001); using reminders (*p* = 0.002); changing the environment (*p* = 0.01); and repeating the behavior (*p* = 0.003) showed statistically significant increases in use.

#### 3.2.3. Participant Behavior: Pre- and Post-Training Patient Data

Patient reports evaluated the participants’ use of communication and behavior change skills in consultations. Patients reported that professionals used fewer communication skills (columns 1–5 in Figure 5 below) and slightly more reminders, support, and goal-setting behavior change techniques (columns 6–12) at follow-up (Time 3) than prior to training (Time 1). One-sample t-tests revealed significant reductions in all communication skills (all *p* = 0.01 or above) at Time 2. Use of the BCTs “identify behavior” and “pros and cons” also significantly reduced (*p* = 0.02; *p* = 0.001, respectively), while there was no significant change in the other BCTs.

### 3.3. Mentoring

Four health and care professionals participated in the two-session mentoring training in St Helena and agreed to mentor others. The role of mentors was to meet regularly with mentees and provide information and support for their peers. Our intention was to also offer regular remote mentoring support during the period post-training to help ensure local sustainability and to support the mentors in their role. One online session was conducted 4 weeks post-training, aiming to support the organization of local mentoring sessions and development of a coaching network, and local mentoring sessions were held in the weeks post-training; however, further planned sessions were not possible due to the imposition of COVID-19 restrictions.

## 4. Discussion

The behavior change training program was well received by St Helena staff and considered to be acceptable and important as part of behavior change consultations with patients. The program demonstrated effectiveness through the increase in facilitators of behavior change, including knowledge, skills, confidence, importance, and future intention to use these techniques. Importantly, there were also reported increases in participant behaviors in relation to the use of the selected BCTs and communication skills, which was maintained at follow-up. These findings are broadly in line with previous similar behavior change training programs that have demonstrated post-course changes in confidence and importance as well as increased use of BCTs [30,33,42,43].

There was a reduction in patient-reported observations of communication skills and BCTs between pre-training and follow-up. There are several reasons why this might be the case. The follow-up data were collected long after the training program was completed, with COVID-19 restrictions having taken place in the interim (and some restrictions were ongoing at the time of the follow-up data collection point). It may be that health practitioners were overwhelmed by the events of the previous two years and were even still focusing primarily on urging patients to follow safety protocols to keep people on the island safe. In another study of health behavior change counseling undertaken during COVID restrictions, the increased pressures on staff interfered with their perceived ability to use behavior change interventions opportunistically [44]. Also, we know that behavior is notoriously difficult to change [7] and clinician behavior no less so [45]. It is reasonable therefore that in the absence of the mentoring program alongside the prolonged period of the pandemic, confidence and the use of techniques may have declined and clinicians simply were not opportunistically using the behavior change skills under these circumstances.

Changing health behaviors in areas as remote as St Helena, where the topology is challenging, requires a behavioral repertoire that accommodates this unique location. The steep hills of St Helena make it difficult to engage in regular daily activities of walking or running, the rocky terrain undermines the capacity for substantial vegetable and fruit planting, and seasonal shortages in some produce are a fact of life for residents. This program acknowledged these challenges and was explicitly adapted to address the issues specific to St Helena through using adapted role play scenarios and opportunities to discuss problems and solutions specific to the area. In other circumstances of behavior change training delivery to remote areas, video platforms have been used effectively. Nevertheless, it appears that training through video conferencing is more likely to result in changes in confidence, knowledge, and skills, and less likely to demonstrate changes in behavior [46,47]. Practicing skills is recommended as a method to increase skill development and implementation, with a higher likelihood that this might have a cascading impact on patient health outcomes [47]. Nevertheless, at the time, St Helena was poorly served by reliable broadband and telecommunication, rendering distance support a challenge.

This whole-system approach to upskilling teams in health behavior change and communication skills has much potential in this type of remote context. This was evidenced by the wide range of professional roles of attendees at the MAP training, including many nurses, clinicians, and public health practitioners; the spirit of local collaboration; and the sharing of knowledge and best practices between those with different roles in health and social care. Local support for the delivery of this intervention was extremely positive. Attendance at each of the sessions was good and there was strong commitment to the training, as evidenced by engagement with the training content, methods, and tasks of the workshops. The opportunity to be able to deliver this training face-to-face undoubtedly had benefits in terms of the level of engagement of the participants and providing useful experience and knowledge for the intervention team to support the tailoring of the program for local use.

Initial informal feedback post-intervention was also very positive. The senior management team was very enthusiastic about the intervention and offered support for the mentoring and coaching program going forward. Two members of staff volunteered to lead on this initially, recruiting two other mentors/coaches from the cohort who participated in training.

Unfortunately, these plans were disrupted by the impact of the COVID-19 pandemic. St Helena normally has a very high turnover of staff, with many coming from different countries committing to a two-year contract. This turnover combined with the COVID pandemic resulted in a loss of some key people who had championed the training and supported the mentoring of other staff, due to moving away from the island or being redirected towards COVID management work. Some of this momentum and enthusiasm was difficult to recover, despite the very strong top–down support.

It is also important to acknowledge some of the methodological flaws of this project. We had originally intended to measure skill competence. Measuring competence can be a methodological challenge in consultation communication practice [48]. It can be difficult to ascertain sustained skill attainment practiced in different scenarios; the process can be time-consuming and potentially overwhelming for some practitioners who are not accustomed to these forms of measurement [49]. This concern was borne out in the acceptability evaluation, where some participants stressed the need for fewer questionnaires. The practicalities of time also meant that we would only be able to assess skill competence immediately post-training, but this would be when skills would still be fresh and not embedded in consultation practice.

Further, there were a relatively small number of participants, and we measured many different variables, suggesting that overall confidence in the statistical power of results may be low. There is always a balance to be struck in terms of how to encourage practitioners onto a training program that might help to facilitate initial changes in consultation practice. Those who are less confident may be discouraged from attending or have a lower perceived need for these types of skills.

We were fortunate to be able to obtain triangulating feedback data from patients to inform our evaluation of participant skills before and after the training. However, these patient participants were volunteers who may have been positive or amenable to taking part, skewing the sample. Since the patient questionnaire was anonymous, we were not able to recruit the same people at follow-up, and initial sample sizes were very uneven, compromising the reliability of the data.

Nevertheless, there is some evidence that the training we offered had some lasting effect in terms of upskilling staff.

## 5. Conclusions

Health behavior change training informed by the MAP framework shows promise in changing health and care practitioner consultation skills to facilitate change in patient behavior. These consultation changes can be detected by patients. In extremely remote areas where health challenges can have a substantial burden on personal, economic and social capital, interventions that can maximise health as well as the self-management of health conditions can be vital.

The training was designed to be cascaded to other practitioners but the infrastructure and consistent transition of staff off and onto the island, alongside restrictions due to the COVID-19 pandemic prevented this percolation of training. With the prospect of improved technological infrastructure on the island, it is possible that future training interventions may lead to sustained changes across the health and social care system, with a positive impact on individuals who access services. Future interventions should also seek to establish the impact on patient health behavior.

## Figures and Tables

**Figure 1 healthcare-13-00435-f001:**
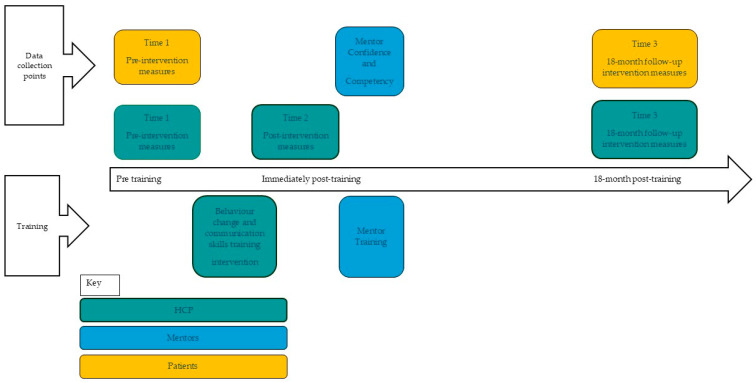
Timeline of intervention and data collection.

**Figure 2 healthcare-13-00435-f002:**
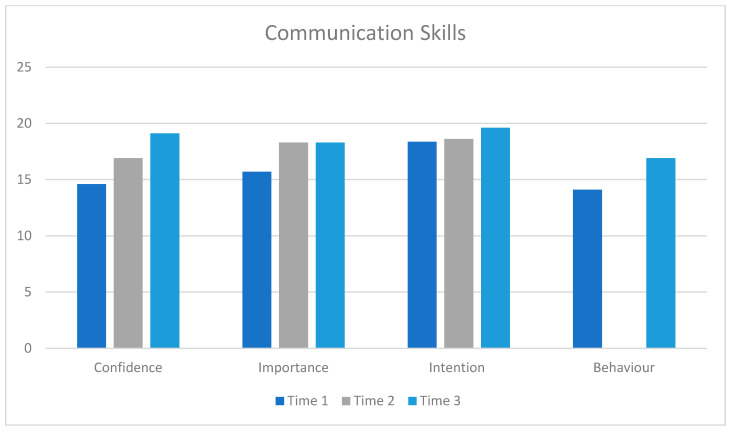
Participants’ self-reported confidence, importance, intention, and behavior for communication skills, comparing Time 1 (3 weeks pre-training), Time 2 (post-MAP training), and Time 3 (18-month follow-up). Post hoc tests: confidence: 1 < 2: *p* = 0.014; 1 < 3: *p* = 0.001; 2 < 3: *p* = 0.008; importance: 1 < 2: *p* = 0.04; 1 < 3: *p* = 0.013; intention: 2 < 3: *p* = 0.008; behavior: 1 < 3; *p* = 0.03. NB: Behavior was measured at Time 1 and Time 3 as participants had no opportunity to implement skills in their practice settings immediately post-training (Time 2).

**Figure 3 healthcare-13-00435-f003:**
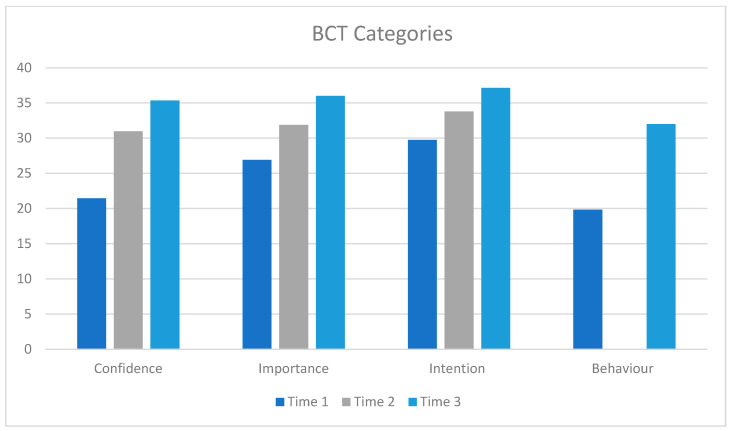
Participants’ self-reported confidence, importance, intention, and behavior for MAP BCTs, comparing Time 1 (3 weeks pre-training), Time 2 (post-MAP training), and Time 3 (18-month follow-up). Post hoc tests: confidence: 1 < 2: *p* < 0.001; 1 < 3: *p* < 0.001; 2 < 3: *p* = 0.004; importance: 1 < 2: *p* < 0.001; 1 < 3: *p* = 0.002; 2 < 3: *p* = 0.04; intention: 1 < 3: *p* = 0.03; 2 < 3: *p* < 0.04; behavior: 1 < 3: *p* = 0.005.

**Figure 4 healthcare-13-00435-f004:**
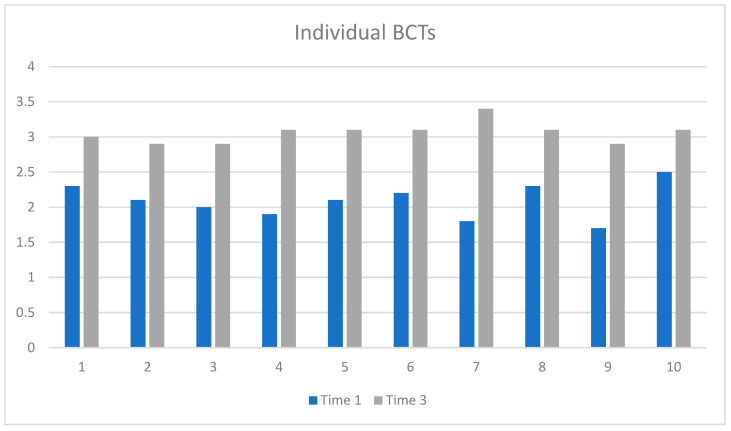
Participants’ self-reported behavior (use of MAP BCTs) comparing Time 1 (3 weeks pre-training) and Time 3 (18-month follow-up). Key: 1, identify behavior; 2, pros and cons; 3, create plan; 4, self-monitor; 4, social support; 6, problem solve; 7, set reminders; 8, change environment; 9, repeat behavior; 10, review progress.

**Figure 5 healthcare-13-00435-f005:**
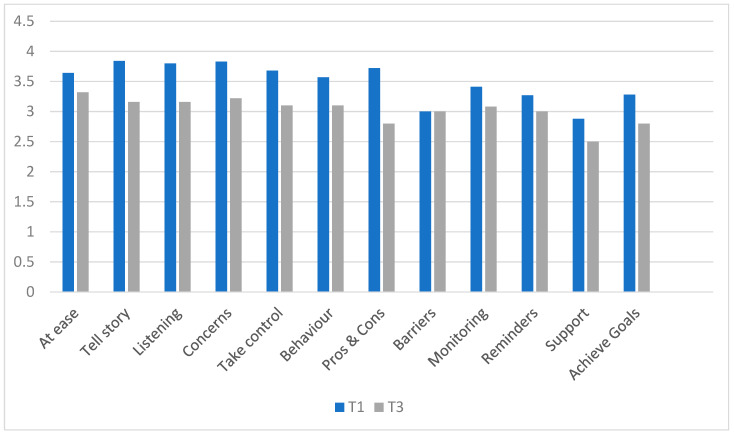
Primary care patient mean ratings of health professionals’ use of behavior change and communication skills at Time 1 (prior to MAP training) and at Time 3 (long-term follow-up).

**Table 1 healthcare-13-00435-t001:** Data collection measures, constructs, Kirkpatrick model and participant group.

Measures	Kirkpatrick Model	Constructs	Effectiveness, Acceptability, Sustainability		Participant Group
BCT and communication questionnaire	Behavior	BCT use	Effectiveness (time 1 & time 2)		1
	Learning	BCT intention			1
	Learning	BCT importance			1
	Learning	BCT confidence			1
	Behavior	Communication skill use		Sustainability (time 1, 2 and 3)	1
	Learning	Communication skill intention			1
	Learning	Communication skill importance			1
	Learning	Communication skill confidence			1
Care MEASURE + BCT/Comm Skill use	Behavior	Patient perception of HCP BCT and comm skills use in consultation	Effectiveness (time 1 & 2)		2
				Sustainability (time 1, 2 and 3)	
TARS	Reaction	Acceptability	Acceptability		1
EMCC mentor skill competency	Behavior	Mentor skill use	Sustainability		1 (mentor subgroup)
EMCC mentor skill competency	Learning	Mentor skill confidence	Sustainability		1 (mentor subgroup)
Key: 1 = HCP training participants 2= Patients of HCPs					
	Behavior				
	Learning				
	Reaction				

## Data Availability

Data are available on request. Please contact the corresponding author.

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
