# Peer review of "An Evaluation of Health Behavior Change Training for Health and Care Professionals in St. Helena"

_healthcare, 2025, doi:10.3390/healthcare13040435_

Round 1
Reviewer 1 Report (Previous Reviewer 1)
Comments and Suggestions for Authors
Thank you for the revised version of the manuscript An evaluation of health behaviour change training for health and care professionals in St. Helena
The manuscript has been rewritten and has improved substantially. However there are still some unclarity that needs to be improved.
On page 2, starting on line 58 the study rationale and aim is now very clearly stated. That is crucial for the reasoning through out the manuscript. Good. – Parts of this could very well be added in the abstract to improve the clarity of the study even there.
P 2 line 82. It is stated that the heal promotion strategy presented “… an important opportunity … “. I wounder if need for is a better wording since the reasoning regards the need for health care professionals to be trained …
On p 3 starting on line137, In the research questions it needs to be a clearer discriminating the participants, patients and practitioners. For example, Rq 2, the participants are mentioned, and it is still not obvious that it means the clients who will be taking part of the trained practitioners’ skills. This needs to be better clarified.
The design described starting on p 4, is still complicated. It is clearer but maybe a figure describing the different sources for intervention and data collection respectively could help.
On p 5, the last paragraph under Delivery regards data collection and I suggest it is moved to that section.
P 5, 2.5 Participants. This section could benefit from a better structure. It should be clarified that participants were recruited for different purposes and at different timepoints. This is not clear. The paragraph starts with describing the inclusion of participants for the “Training intervention”. The paragraph is presenting the follow up. New participants, and “the patients are recruited …” is confusing. Again, if there were a figure illustrating the study design and sources for data collection it could be referred to here and maybe it becomes more obvious how the different groups are related.
In the same paragraph strategies for data collection are mentioned. I suggest this should be moved to 2.6 Evaluation measures.
The table added to this paragraph shows good intentions. However, I find it hard to follow. I am also confused about the added points below; 1 and 2 followed by a and b. Maybe it is due to the added table in the manuscript but are these points meant to be sub-notes to the table? It is also confusing that the added points are followed by a paragraph, p 7 starting on line 286 that seems to describe steps of analysis.
P 7 line 295. How does the heading 2.7 discriminate to the previous heading evaluation measures? Should these two headings be two subheadings under Data collection?
The paragraph under Analyses on p 7 starting om line 323, describes and argues for the use of open-ended questions. This needs to be rewritten. Under analysis there should be a description of how the questions were analysed but to what instrument they were added or when/how the data were collected should be moved to a section regarding data collection.
On p 7 line 337 I believe the v.s. should be capital letters? (The same error (?), is repeated under the note Authors' contributions, at the end of the manuscript.)
The presentation of the Results, is hard to follow. Maybe it is because of the layout in the revised manuscript but still it could benefit from improved structure.
Maybe a short introduction describing how the result is presented could be added. Now the whole section starts with just “Engagement:”, p 8 line 344. The number of all participants should be added to complement the n=4 information, on line345.
Course evaluation: (p 8 line 347).
On p 8, line 351 the sentence starts with “30”. This should be written in letters and maybe “…of 32 participants” should be added.
On p 8 line 357 the sentence “The open responses were then categorized into three central themes…”. Is there a mutual question that is answered by these themes? The themes are now separate, and it is not clear if and how they contribute to an overall understanding? – I suggest the themes are presented in a similar wording, to get a mutual understanding of … what?
The problem is illustrated by the theme presented on p 8 line 368; ii) In their new understanding of the value of collaborative consultations: - What does this them mean? Who is referred to? What is happening?
I also suggest that the sentence (line 357) “The open responses were then categorized into three central themes…” could be supplemented with “… illustrated by examples below”. This way the “For example:” – texts (line 364, 369 and374) are not needed and may improve the reading of the text.
P 9 line 378, the headings and subheadings in the result-section is unclear. Is this a subheading of 3. Results? This needs to be adjusted for the whole section.
Two figures are presented but it is unclear which one is the one referred as Figure 1, or both?
On p 10, line 404 and p9, line 394 a similar note is presented. Are both two figures meant to be in the manuscript?
It would be beneficial if the statistically significant changes were marked in the figures.
I find the discussion section to be relevant and as I mentioned the first time the study is important and interesting, but it needs to be presented in a way that makes the discussion relevant. There are several good points made in the discussion that needs to be better supported by a clear method and result section.
Author Response
Please see attachment

Reviewer 2 Report (Previous Reviewer 2)
Comments and Suggestions for Authors
The manuscript is much easier to follow now. There are a few very minor items to be addressed:
Line 101: Typo: BCTs (remove apostrophe)
Line 115: The programmed draws on 10 BCTs from the MAP framework....should that be from the HBCCF framework?
Line 235: Grammar: Data were collected
Line 281: Sentence not complete.
Line 317: Grammar: Missing data were replaced
Line 319: Grammar: Data pre- and post- were compared
Line 394: Grammar: Data at time 2 for behaviour were not collected
Author Response
Please see attached.

Reviewer 3 Report (Previous Reviewer 4)
Comments and Suggestions for Authors
The authors revised the manuscript to my satisfaction because they addressed the comments provided.

Author Response
We understand that you had no advise regarding corrections, however I provide a summary of the comments and suggestions made by the other reviewers and our response to them for your information. Please see attached. And likewise, we would like to thank you for your time and effort in providing a review.

This manuscript is a resubmission of an earlier submission. The following is a list of the peer review reports and author responses from that submission.
Round 1
Reviewer 1 Report
Comments and Suggestions for Authors
Thank you for the opportunity to review your manuscript. I understand the need for the intervention, and it is evident that the Covid pandemic greatly impacted the study design. Despite this, there are comments and questions regarding the manuscript that need major revisions and clarifications.
The title is hard to understand. Maybe there are some words missing. “… Training for Multidisciplinary Healthcare …”. I think adding healthcare professionals or staff would help.
The aim is described differently in the abstract (line 15) and at the end of the introduction (line 87). This needs to be clarified and corrected.
In the aim presented as following the introduction they introduce terms as feasibility, effectiveness, acceptability and sustainability (line 87-89) These terms are not touched upon earlier in the introduction, in the method section or the discussion. This makes the aim somewhat separated from the rest of the study and needs to be handled.
In research question 4, (line 96) it is unclear what participants are regarded? The participants in the intervention or patients receiving BCT from the participants who had gotten the intervention?
The method section
I find this section rather confusing, and it is hard to see how all the different assessment used, are motivated.
The section Delivery (starting on line 128) could be improved by a figure describing the longitudinal characteristics of the study. Also, the TIME 1, 2 and 3 used in the result section could be introduced in such figure.
On line 135 observations, audio and video recordings are mentioned but this is not presented in the result section. How was this data handled?
On line 138 the identification of four mentors is introduced. This was not mentioned in the study design, and it is unclear if this strategy was part of the actual study?
The participants (Line 156) could be described in a table to improve readability.
Regarding the pre and post intervention triangulation design, collecting data from patients (line 172) it is unclear how the samples of 25 and later 74 patients were matched regarding age and sex. Why did you decide to collect data from three times more patients after the intervention?
On line 177 the evaluation measures are presented. Here you introduce the Kirkpatrick’s hierarchical model. How does this relate to the terms of feasibility, effectiveness, acceptability, and sustainability that were supposed to be assessed according to the expression in the aim?
The description of the different measures used is hard to follow. A table of the measures used and when, could be useful.
Line 211-214 is not related to the specific instruments and could better move to a timeline showing the study design.
The result section.
In figure 1, why is a bar lacking for Behaviour T2?
The credibility of figure 1, 2 and 3 could be improved by indicating what differences that were statistically significant, between T1, T2 and, T3.
Starting on line 305 the patients’ perception of the participants' behavior is illustrated. It is unclear what figure 4 adds. It seems like a more descriptive illustration. Why are the results of the t-tests not presented?
Why is the paragraph starting on line 314, regarding the mentoring presented as a result? It was not part of the aim or not related to any of the research questions. Could this part be mentioned in the discussion instead?
The discussion section
You are very transparent in discussing your results which is a strength. However, several of the limitations are not regarded and some could have been avoided in the study design process.
References
The references seem to be presented in different formats.
Reviewer 2 Report
Comments and Suggestions for Authors
This is a behaviour change intervention on a small scale in St. Helena that was partially interrupted by the Covid-19 pandemic.
The MAP framework is used to guide the intervention and the two broad areas of focus were to address:
· BC communication skills (knowledge, competence, confidence)
· use of BC techniques (knowledge, competence (confidence is missing)).
Throughout the document: please confirm if you mean “interdisciplinary” rather than multidisciplinary”, as multidisciplinary refers to collaboration that maintains disciplinary boundaries, whereas interdisciplinary involves developing new approaches and perspectives by blending those disciplinary boundaries.
Could the key research questions be stated more clearly? It is not clear, for each question, who is being referred to. Also, the key research questions do not match the previously stated broad areas of focus. Specifically, regarding BC communication skills, there is no mention of knowledge of those skills; similarly for use of BC techniques (BCT) there is no mention of knowledge).
Explain why “intention to use” is being measured when actual use is also being measured. What is the advantage of measuring the former?
However, assuming the key research questions remain as stated, there are actually several questions being posed:
1. Can a health behavior change co-developed training intervention increase confidence in use of BCT?
2. Can a health behavior change co-developed training intervention increase confidence in BC communication skills?
3. Can a health behavior change co-developed training intervention increase importance (should this item be “perceived importance”?) in use of BCT?
4. Can a health behavior change co-developed training intervention increase importance (should this item be “perceived importance”?) in BC communication skills?
5. Can a health behavior change co-developed training intervention increase intention (should this item be “intention to use”?) of BCT?
6. Can a health behavior change co-developed training intervention increase importance (should this item be “intention to use”?) of BC communication skills?
7. Can a health behavior change co-developed training intervention increase use of BCT?
8. Can a health behavior change co-developed training intervention increase use of BC communication skills?
9. Can this intervention increase competence in BC communication?
10. Can this intervention increase competence in BCT use?
11. Will there be a change in patient perception of practitioner BC communication post training?
12. Will there be a change in patient perception of practitioner BCT skills post training?
13. What are participants’ perceptions of the training intervention?
However, it does not appear that the competence in BC communication and competence in BCT use have been measured.
There were only 25 pre-intervention participants; it is not clear whether the 74 post intervention participants included the original 25. It is therefore a bit of a stretch to compare patients’ data, especially with such a small sample size.
Lines:
152: what does the fact that only 32/250 eligible staff were willing and available to attend training mean in terms of uptake and sustainability of the intervention?
164: Were the 14 respondents part of the original 32?
164: Does “long-term follow-up” refer to the 18 month post-intervention time?173: Post-intervention data from patients: at what time were those data collected?
Reviewer 3 Report
Comments and Suggestions for Authors
The authors present a behavioral change training program for residents of St. Helena, a small island in the south Atlantic. I was impressed by the planning and execution of this intervention program, including modifying the contents of the programme to fit the cultural and historical context of the St. Helena residents and sending members of the research team to the island to conduct the training program in person. If not for the COVID-19 pandemic occurring and preventing the team from returning to the island, I think the results from data would have been more impressive.
Overall, I think this manuscript would be a good contribution to this journal, but I have two comments for improvement:
1) I would like to see a little more explanation of BCT and MAP framework in the Introduction section. For example, the authors mentioned that BCT's can be categorized by their function, but how many categories are there? Also, how are these different BCT's related to each other?
2) Relatedly, I'd also like to see a little explanation about the evaluation measures in the Methods section. For example, the authors stated that they assessed 10 different BCT's, suggesting that there are at least 10 different categories (see point 1). Why were these 10 selected? Could the authors provide examples of the specific statements/questions that were used for this assessment? Additionally, could the authors explain their rationale for also including the communication skills measures from the CARE and how this scale is theoretically related to MAP and BCTs, if at all.
Reviewer 4 Report
Comments and Suggestions for Authors
The authors evaluated the impact of health behavior change. This study is relevant for health promotion and enable people to develop self-management and health outcomes.
